# Diverse microtubule-targeted anticancer agents kill cells by inducing chromosome missegregation on multipolar spindles

**Amber S. Zhou[1], John B. Tucker[2], Christina M. Scribano[1], Andrew R. Lynch[3], Caleb L. Carlsen[4], Sophia T. Pop-Vicas[5], Srishrika M. Pattaswamy[5], Mark E. Burkard[6,7,8], Beth A. Weaver[5,7,8]***

1 Molecular and Cellular Pharmacology Graduate Training Program, University of Wisconsin, Madison, Wisconsin, United States of America, 2 Cancer Biology Graduate Training Program, University of Wisconsin, Madison, Wisconsin, United States of America, 3 Cellular and Molecular Pathology Graduate Training Program, University of Wisconsin, Madison, Wisconsin, United States of America, 4 Cellular and Molecular Biology Graduate Training Program, University of Wisconsin, Madison, Wisconsin, United States of America, 5 Department of Cell and Regenerative Biology, University of Wisconsin, Madison, Wisconsin, United States of America, 6 Department of Medicine, University of Wisconsin, Madison, Wisconsin, United States of America, 7 Department of Oncology/McArdle Laboratory for Cancer Research, University of Wisconsin, Madison, Wisconsin, United States of America, 8 Carbone Cancer Center, University of Wisconsin, Madison, Wisconsin, United States of America

* baweaver@wisc.edu

**Data Availability Statement:** All relevant data are within the paper and its Supporting Information files.

## Abstract

Microtubule-targeted agents are commonly used for cancer treatment, though many patients do not benefit. Microtubule-targeted drugs were assumed to elicit anticancer activity via mitotic arrest because they cause cell death following mitotic arrest in cell culture. However, we recently demonstrated that intratumoral paclitaxel concentrations are insufficient to induce mitotic arrest and rather induce chromosomal instability (CIN) via multipolar mitotic spindles. Here, we show in metastatic breast cancer and relevant human cellular models that this mechanism is conserved among clinically useful microtubule poisons. While multipolar divisions typically produce inviable progeny, multipolar spindles can be focused into near-normal bipolar spindles at any stage of mitosis. Using a novel method to quantify the rate of CIN, we demonstrate that cell death positively correlates with net loss of DNA. Spindle focusing decreases CIN and causes resistance to diverse microtubule poisons, which can be counteracted by addition of a drug that increases CIN without affecting spindle polarity. These results demonstrate conserved mechanisms of action and resistance for diverse microtubule-targeted agents.

**Trial registration:** clinicaltrials.gov, NCT03393741.

## Introduction

Microtubule poisons are one of the most commonly used therapies for numerous malignancies, including primary and metastatic breast cancers of all subtypes [1–3]. Improved understanding of mechanisms that dictate response to microtubule poisons is urgently needed, as a

**Funding:** This work was supported, in part, by National Institutes of Health grants P30 CA014520 (UW Carbone Cancer Center); R01CA234904 (to B. A.W. and M.E.B.), T32 GM008688 (to A.S.Z.), T32 CA009135 (J.B.T, C.M.S), F31CA254247 (to A.R. L), and T32 GM141013 (to C.L.C). The funders had no role in study design, data collection and analysis, decision to publish, or preparation of the manuscript.

**Competing interests:** M.E.B. declares the following: Medical advisory board of Strata Oncology; Research funding from Abbvie, Arcus, Apollomics, Elevation Oncology, Endeavor, Genetech, Puma, and Loxo Oncology, Seagen. All other authors declare that they have no conflict of interest.

**Abbreviations:** CENP-Ei, CENP-E inhibition; CIN, chromosomal instability; dox, doxycycline; NEBD, nuclear envelope breakdown; Plk4, Polo-like kinase 4; shRNA, short hairpin RNA.

substantial proportion of patients derive no benefit from this cornerstone of treatment and suffer needless toxicity and delays in effective treatment. Although the best-selling chemotherapy drug paclitaxel (Taxol) is considered highly effective, only 41% to 58% of patients respond [4]. Similar fractions of metastatic breast cancer patients respond to other microtubule poisons, including docetaxel (30% to 63%), eribulin (12% to 29%), and vinorelbine (15% to 50%) [5–12]. Combination therapies that could sensitize the large number of tumors that are resistant to microtubule poisons would have a profound clinical impact.

Microtubule poisons have traditionally been described as microtubule stabilizers or destabilizers [13,14]. Microtubule stabilizers (paclitaxel, docetaxel) promote polymerization of purified tubulin subunits in vitro and increase microtubule polymer mass in cells [15–17]. Microtubule destabilizers (vinblastine, vinorelbine) inhibit microtubule polymerization in vitro and reduce microtubule mass in cells, with sufficiently high concentrations eliminating mitotic spindles [18,19]. Interestingly, low doses of both stabilizers and destabilizers have very similar effects on microtubules in vitro and suppress microtubule dynamics without affecting polymer mass [20–23]. Though high (μM) concentrations of microtubule stabilizers and destabilizers have opposing effects on microtubule polymer mass in cells, both arrest cells in mitosis due to activation of the mitotic spindle assembly checkpoint [24]. Mitotic arrest results in either mitotic cell death or an aberrant exit from mitosis without anaphase that produces a tetraploid G1 cell, a process termed "adaptation" [25]. Cells that are sensitive to high doses of microtubule poisons either die during mitotic arrest or after mitotic exit. For decades, it was expected that microtubule poisons exert anticancer activity by arresting cells in mitosis [25–27]. This expectation led to the development of novel antimitotic drugs that arrest cells in mitosis without affecting microtubules, including drugs against Eg5/KSP [28–32], Plk1 [33–37], CENP-E [38–40], and Aurora A [41–44]. Though these drugs were largely successful in causing mitotic arrest, unfortunately, they did not show similar efficacy as microtubule-targeted therapies in clinical trials [31,32,36,37,45–47]. The reasons for this were unclear.

More recently, in 2 clinical trials of primary breast cancers treated with both standard-of-care doses of paclitaxel, we directly sampled tumors after treatment and discovered that paclitaxel does not accumulate to the high concentrations necessary to cause mitotic arrest [48,49]. Rather, the low concentrations achieved in patient tumors induce abnormal, multipolar mitotic spindles in every patient tumor examined. When this is recapitulated in the laboratory, timelapse analysis reveals that cells treated with low, clinically relevant concentrations of paclitaxel do not arrest in mitosis but enter anaphase and divide on a mitotic spindle that is at least transiently multipolar [48,49]. This paradigm-shifting discovery revealed a novel anticancer mechanism for paclitaxel and raised the question of whether other clinically useful microtubule poisons induce mitotic arrest or, like paclitaxel, cause abnormal mitotic transit on multipolar spindles.

Aneuploidy, an abnormal chromosome number that differs from a multiple of the haploid, is found in a majority of cancers, often due to ongoing chromosome missegregation, also known as chromosomal instability (CIN) [50–52]. Cancer can tolerate and even benefit from some CIN, as it provides genetic variation for evolution under various selective pressures [50,53–60]. However, high rates of CIN cannot be sustained in cancer [61–68], likely due to complete loss of essential chromosomes and/or stress-induced antiproliferative effects [69–73]. Baseline CIN positively correlates with response to microtubule stabilizing drugs in metastatic breast cancer [49], suggesting that paclitaxel is effective in patient tumors in which it increases CIN over a maximally tolerated threshold. Despite this relationship, the minimum rate of CIN that sensitizes cells to paclitaxel has not been defined.

Here, we demonstrate that a range of diverse clinically useful antimicrotubule drugs, but not the microtubule destabilizers nocodazole or colcemid, which are not used clinically, have a

conserved mechanism of action. Timelapse analysis reveals that diverse microtubule poisons induce a mitotic spindle that starts out multipolar in early mitosis. In cases in which the multipolar spindle is sustained, it results in high CIN and cell death. In other cases, the muItipolar spindles are focused into near-normal bipolar spindles through clustering of extra poles. Quantifying the rate of CIN when >3 chromosomes are missegregated, as occurs in multipolar divisions, is challenging because the missegregated chromosomes do not form discrete masses. We therefore developed a novel method to quantify CIN and demonstrate that cell death positively correlates with net loss of DNA, which often occurs >24 hours after division. In cells that focus multipolar spindles prior to anaphase onset, which typically produce viable progeny, the addition of a drug to increase CIN through another mechanism substantially increases the percentage of daughter cells that lose DNA and induces synergistic lethality with microtubule poisons. Increasing CIN by loss of an additional 3 to 4 chromosomes per diploid genome is sufficient to improve sensitivity to both microtubule stabilizing and destabilizing drugs. Together, these results demonstrate that diverse microtubule poisons induce multipolar spindles early in mitosis. Focusing multipolar spindles into near-normal bipolar spindles represents a conserved mechanism by which cells can reduce DNA loss and evade cell death. Further, this mechanism of resistance to antimicrotubule agents can be overcome by combination treatment with a second CIN-inducing drug.

## Results

### Clinically useful microtubule poisons induce multipolar spindles

To determine whether the clinically relevant mechanism of paclitaxel is conserved among microtubule poisons, we tested whether clinically useful antimicrotubule agents are capable of inducing multipolar spindles without mitotic arrest. Because low nM concentrations of paclitaxel generate clinically relevant intracellular concentrations, we tested whether comparable concentrations of other microtubule poisons exerted similar effects in cell culture. Triple negative (negative for expression of estrogen receptor, progesterone receptor, and overexpression of HER2) breast cancer cell lines dervied from metastatic cancer (Cal51 and MDA-MB-231) were treated with increasing concentrations of microtubule poisons classically considered to be stabilizers (docetaxel, ixabepilone, epothilone B) or destabilizers (vinblastine, vinorelbine, eribulin). All were capable of inducing multipolar spindles very similar to those caused by paclitaxel at low (0.1 to 10) nM doses (Fig 1A–1D and Fig A in S1 File). The lowest concentrations that induced multipolar spindles did so without causing the large increase in mitotic index produced by high doses of drug, which cause mitotic arrest (Fig B in S1 File). Interestingly, the microtubule destabilizing drugs nocodazole and colcemid, which are not used clinically, did not induce multipolar spindles (Fig C in S1 File). Thus, clinically useful microtubule poisons are capable of inducing multipolar spindles without mitotic arrest in cultured cell models.

To determine whether microtubule poisons other than paclitaxel cause multipolar spindles without mitotic arrest in patient tumors, we obtained tumor biopsies from female patients with metastatic breast cancer before and after 20 hours of treatment with single-agent antimicrotubule therapy. The trial design is shown in Fig 1E. Patient characteristics are in Table A in S1 File. Prior to therapy, most mitotic cells observed in patients' core biopsy sections exhibited an apparently normal bipolar mitotic spindle (Fig 1F top, and 1G). About 20 hours of treatment with vinorelbine, eribulin, or nab-paclitaxel increased the incidence of multipolar spindles without substantially affecting mitotic index (Fig 1F bottom, 1G and 1H). These data provide a clinical benchmark that supports a conserved mechanism among clinically relevant microtubule poisons in which they induce multipolar spindles without mitotic arrest.

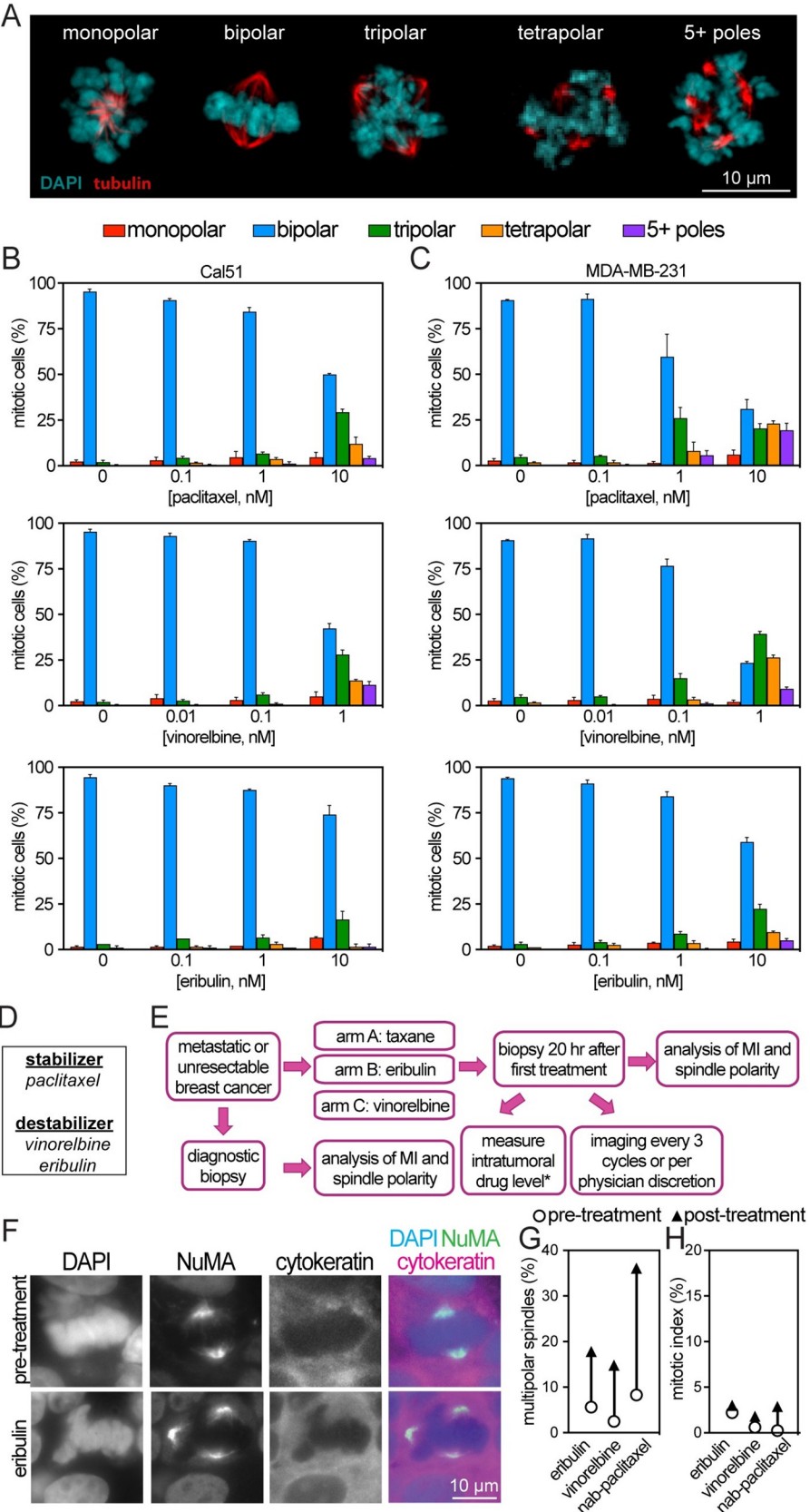

**Fig 1. Clinically useful microtubule poisons induce multipolar spindles.** (**A**) Representative images of mitotic cells with the indicated spindle polarity in Cal51 cells treated with 10 nM paclitaxel. (**B, C**) Polarity of mitotic spindles from all stages of mitosis in Cal51 (**B**) and MDA-MB-231 (**C**) cells treated with the indicated microtubule poison for 20 hours. Data indicate mean +/− SE. $n \geq 100$ cells per condition in each of 3 biological replicates. (**D**) Table of microtubule stabilizers and destabilizers. (**E**) Schematic of clinical trial treatment. Patient characteristics appear in Table A in S1 File. MI, mitotic index; MT, microtubule. *, eribulin only; see Table B in S1 File. (**F**) Representative images of mitotic cells from patient biopsies. (**G, H**) Quantification of multipolar spindles (**G**) and mitotic index (**H**) in patient biopsies before (open circles) and 20 hours after (triangles) treatment with the indicated microtubule poison. (**G**) $n = 107$, 40, and 12 cells pretreatment and 73, 54, and 122 cells after treatment, respectively, from single patients treated with eribulin, vinorelbine, or nab-paclitaxel. (**H**). $n \geq 300$ cells per condition. Data used to generate graphs can be found in S1 Data.

## Centrosome amplification can promote multipolar spindle maintenance and daughter cell death in diverse microtubule poisons

If other microtubule poisons have a mechanism of action similar to paclitaxel, we expect them to show similar sensitivities. We previously showed that centrosome amplification due to doxycycline (dox)-inducible Polo-like kinase 4 (Plk4) overexpression in MCF10A cells [74] increases multipolar spindle maintenance and daughter cell death after treatment with a sub-clinical dose of the microtubule stabilizing drug paclitaxel [49]. We tested whether Plk4 over-expression had a similar impact after treatment with the microtubule destabilizing drug vinorelbine by performing 72-hour timelapse microscopy of MCF10A cells inducibly express-ing Plk4 [74] and stably expressing histone H2B-mNeonGreen and mScarlet-tubulin to label chromosomes and microtubules, respectively. Approximately 50% of fluorescently labeled MCF10A cells exhibited centriole amplification following 72-hour dox treatment, consistent with Plk4 overexpression (panels A-D of Fig D in S1 File). We selected doses of paclitaxel and vinorelbine that were insufficient to cause a substantial increase in multipolar divisions when used alone (Fig 2A). Plk4 overexpression markedly increased the incidence of multipolar mitotic spindles in early stages of mitosis (prometaphase and metaphase; Fig 2A), though these often focused into bipolar spindles prior to anaphase onset (Fig 2B). Addition of microtubule poisons did not significantly affect the incidence of multipolarity in early stages of mitosis as compared to Plk4 overexpression alone (Fig 2A). However, combining centriole amplification with microtuble poisons considerably reduced the amount of multipolar spindle focusing observed in cells without microtubule poisons, and most MCF10A cells overexpressing Plk4 treated with a microtubule poison had a multipolar spindle at anaphase onset (Fig 2B). Along with the substantial increase in multipolar divisions, Plk4 overexpression dramatically increased cell death in paclitaxel and vinorelbine (Fig 2C). Thus, low concentrations of pacli-taxel and vinorelbine exhibit similar response and sensitivity to Plk4 overexpression in MCF10A cells.

Further examination of timelapse analysis revealed that cells exhibited 4 types of mitotic divisions. We designated these based on (1) spindle polarity at prometaphase; (2) spindle polarity at anaphase onset; and (3) the number of daughter cells formed. Control cells typically maintained a bipolar spindle throughout mitosis and formed 2 daughter cells, which we desig-nated 2->2->2 (Fig 2D top, S1 Movie). Some cells exhibited a multipolar spindle early in mitosis but rapidly focused this into an apparently normal bipolar spindle before anaphase onset. Many of these completely aligned their chromosomes before segregating in 2 directions and forming 2 daughter cells ($3^+$->2->2; Fig 2D second row, S2 Movie). Other cells main-tained a multipolar spindle at least until anaphase onset but ultimately formed 2 daughter cells either because they focused their spindle in anaphase or telophase or because of partial cytoki-nesis failure ($3^+$->$3^+$->2; Fig 2D, third row, S3 Movie). Lastly, there were cells that maintained a multipolar spindle throughout mitosis and formed 3 or more daughter cells ($3^+$->$3^+$->$3^+$;

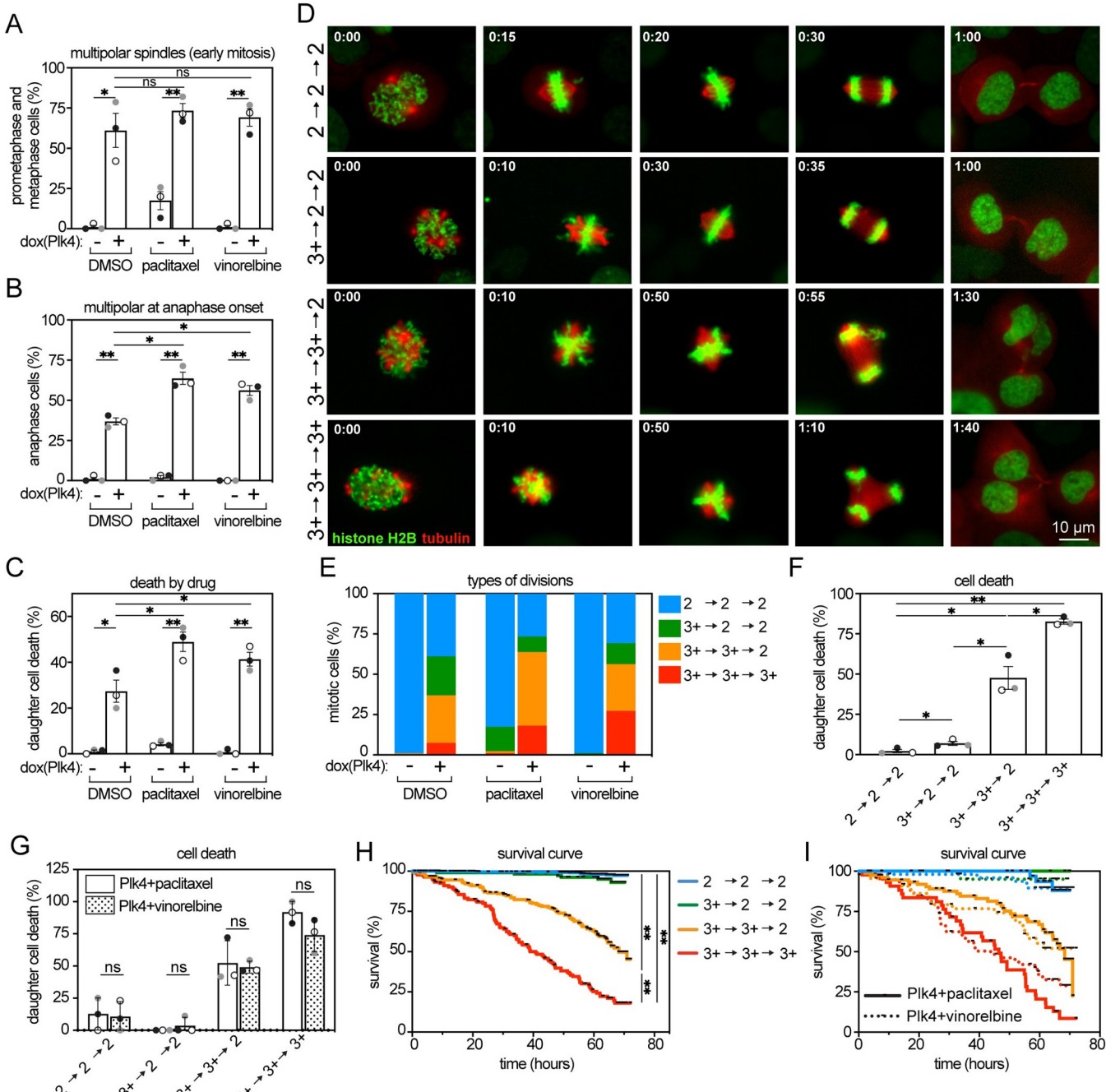

**Fig 2. Centrosome amplification sensitizes to diverse microtubule poisons by increasing multipolar divisions.** Data from 72-hour timelapse imaging of Plk4-inducible MCF10A cells stably expressing histone H2B-mNeonGreen and mScarlet-tubulin treated with 1 nM paclitaxel or 1 nM vinorelbine +/− 2 μg/mL dox to induce Plk4 expression. (**A**, **B**) In cells treated with these low concentrations of microtubule poisons that did not substantially induce multipolar spindles on their own, dox-inducible Plk4 expression increased the incidence of multipolarity prior to (**A**; prometaphase and metaphase) and at (**B**) anaphase onset. $n \geq 80$ cells per condition across 3 replicates. (**C**) Increasing multipolarity substantially increased cell death. $n \geq 147$ cells per condition across 3 replicates. (**D**) Representative still frames of the specified type of division (also shown in S1–S4 Movies), which were characterized based on (1) spindle polarity at prometaphase and metaphase; (2) spindle polarity at anaphase onset; and (3) the number of daughter cells formed. Normal divisions are indicated by 2->2->2. Spindle multipolarity and divisions that produce 3 or more daughter cells are indicated by $3^+$. Numbers in upper left corners indicate time in hours:minutes. (**E**) Quantification of the frequency with which specific types of divisions occurred. $n \geq 80$ cells per condition across 3 replicates. (**F**, **G**) Quantification of cell death after the specified type of division showing that persistent multipolar ($3^+$->$3^+$->$3^+$) divisions are the most lethal across all treatment conditions (**F**) and that this is the case irrespective of which microtubule poison is used (**G**). F includes cells from all treatment groups (including DMSO, $n \geq 175$ cells in each of 3 replicates), while G only includes cells from the specified treatment condition ($n \geq 50$ cells in each of 3 replicates). (**H**, **I**) Kaplan–Meier survival curves of cells with specified types of divisions showing that daughter cells die most rapidly after a persistent multipolar division, irrespective of which microtubule poison

was used. Cell death often required >24 hours. Black points along curves indicate censored cells, which left the field of view or entered a second mitosis. $n \geq 200, 22, 58,$ and 39 cells for 2->2->2, $3^+$->2->2, $3^+$->$3^+$->2, and $3^+$->$3^+$->$3^+$, respectively, in each of 3 biological replicates in H. $n = 44, 16, 76,$ and 45 cells for 2->2->2, $3^+$->2->2, $3^+$->$3^+$->2, and $3^+$->$3^+$->$3^+$, respectively, in Plk4+paclitaxel and $n = 51, 20, 46,$ and 62 for 2->2->2, $3^+$->2->2, $3^+$->$3^+$->2, and $3^+$->$3^+$->$3^+$, respectively, in Plk4+vinorelbine across 3 biological replicates. Color represents specific replicate and bars represent mean +/− SEM for panels A-C, F and G. Unpaired $t$ test was performed on A-C, F and G. Log-rank (Mantel–Cox) test was performed on H to determine statistical significance. Data used to generate graphs can be found in S1 Data. * indicates $p < 0.05$. ** indicates $p < 0.01$. ns indicates not significant.

Fig 2D, fourth row, S4 Movie). While cells treated with these low, subclinical doses of paclitaxel or vinorelbine alone predominantly exhibited 2->2->2 divisions, Plk4 overexpression increased the incidence of $3^+$->$3^+$->$3^+$ and $3^+$->$3^+$->2 divisions in both microtubule poisons (Fig 2E). In both paclitaxel and vinorelbine, $\geq 75\%$ of the daughter cells resulting from $3^+$->$3^+$->$3^+$ divisions died within the 72-hour timelapse analysis (Fig 2F and 2G). Approximately 50% of daughter cells from $3^+$->$3^+$->2 divisions died (Fig 2F and 2G). However, <10% of cells from $3^+$->2->2 divisions died, demonstrating that early focusing of multipolar spindles causes resistance to both paclitaxel and vinorelbine (Fig 2F and 2G). Daughter cells from $3^+$->$3^+$->$3^+$ divisions died substantially more rapidly than cells arising from other types of divisions, whether they occurred in paclitaxel or vinorelbine (Fig 2H and 2I). Interestingly, the timing of death was quite variable, ranging from 1 to 67 hours after division, with an average of approximately 34 hours, suggesting that the timing of cell death is karyotype specific rather than a general response to DNA loss (Fig 2H and 2I). These data demonstrate that Plk4 overexpression sensitizes to both paclitaxel and vinorelbine in MCF10A cells. Additionally, both the frequency and timing of cell death are determined by the type of division rather than the type of microtubule poison, offering further support for a conserved mechanism.

To test whether centrosome amplification promotes multipolar spindle maintenance more generally, we generated Cal51 cells with dox-inducible expression of Plk4. Despite centriole amplification in approximately 40% of cells, Plk4 overexpression only subtly increased multipolar spindles in this cell line, which readily focuses multipolar spindles induced by microtubule poisons (panels A and B of Fig E in S1 File). Unlike in MCF10A cells, treatment with a subclinical dose of paclitaxel did not result in a substantial increase in multipolar spindles in centriole-amplified cells (panels C and D of Fig E in S1 File). Thus, centrosome amplification can, but does not necessarily, induce and promote maintenance of multipolar spindles.

## Histone H2B fluorescence intensity as a proxy for DNA content

Resistance is a major limitation of antimicrotubule therapy and can occur due to focusing of multipolar spindles early in mitosis. Preventing focusing dramatically increases chromosome missegregation and daughter cell death, but unlike for lagging or misaligned chromosomes, it is not possible to quantify the rate of CIN in multipolar divisions by simple inspection, since the missegregated chromosomes in multipolar divisions are not physically separated from the rest of the DNA. We therefore tested whether we could use histone H2B fluorescence intensity as a proxy for DNA content and CIN. We assigned the DNA content of a diploid cell prior to DNA replication as 100%, with the mother cell in mitosis containing 200% DNA (Fig 3A). We considered the DNA content of a mother cell as the product of the mean fluorescence intensity and area of histone H2B signal, which is distributed into the daughter cells (Fig 3A). In a normal division, the DNA content of the mother is divided equally into 2 daughter cells. Consistent with this, divisions in vehicle treated control cells that did not exhibit visible mitotic defects produced daughter cells with 100% DNA content (Fig 3B). To confirm that the DNA content in the mother cell was accounted for in the daughters, histone H2B fluorescence measurements of mother cells 15 minutes prior to nuclear envelope breakdown (NEBD) were compared with histone H2B fluorescence of their daughter cells once their DNA completely

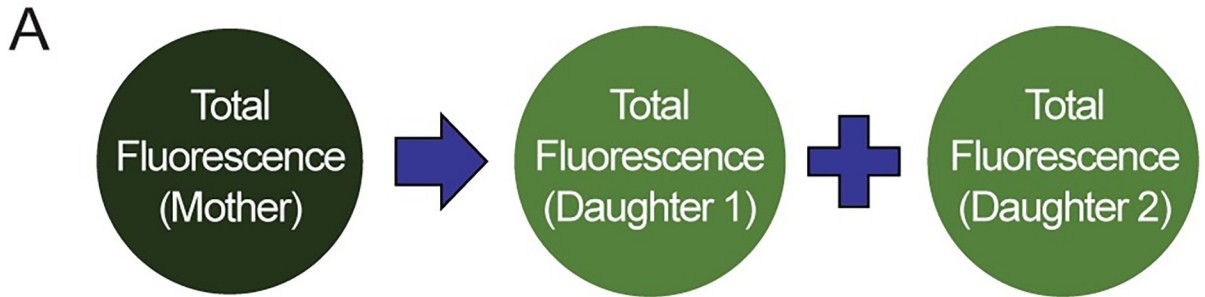

Area * Mean Fluorescence Intensity = Total Fluorescence (TF)

$$\% \text{ DNA Content}_{(Daughter\ 1)} = \frac{TF_{(Daughter\ 1)}}{Sum(TF_{(Daughter\ 1)} + ... + TF_{(Daughter\ X)})} * 200$$

**Fig 3. Validation of histone H2B fluorescence intensity as a proxy for DNA content.** (**A**) Rationale and method for calculating DNA content in a normal division that produces 2 daughter cells (top) and divisions that produce ≥2 daughters (bottom). (**B**) Quantification of DNA content in daughter cells from untreated MCF10A and Cal51 cells without visible segregation errors. $n \geq 60$ cells. (**C**) Quantification of DNA ratio between mother cells in late G2 and summation of daughter cells in early G1, showing mother DNA is accounted for in daughter cells. $n \geq 25$ mother–daughter pairs across 3 biological replicates in MCF10A cells. (**D**) DNA content measurements remain stable in paired daughter cells over 13 hours in MCF10A cells. Data represent 30 daughter cell pairs across 3 biological replicates. (**E**) Quantification of DNA content in single chromosomes of the indicated type in MCF10A cells. Expected DNA content ranges from approximately 0.8% to approximately 4.1% for chromosomes 21 and 1, respectively, in a diploid human female cell. Bars represent mean +/− SEM. Data used to generate graphs can be found in S1 Data.

decondensed in G1. The average ratio of DNA content in the mother:daughters was 0.97–0.98 (Fig 3C), very close to the expected value of 1. To ensure that the measurement of DNA content in the mother cell was consistent over late G2, additional quantifications were performed in mother cells 25 and 50 minutes prior to NEBD. As expected, measurements taken at 15, 25, and 50 minutes prior to NEBD gave consistent results (Fig 3C). Since daughter cells often decondense their DNA at different time points, we tested whether the time point of analysis impacted the results. However, DNA content measurements remained consistent over a 13-hour observation period (Fig 3D), indicating that this analysis is not dependent on the specific time point quantified. To further validate this method, we tested whether, in cases in which an individual chromosome was separated from the remainder of the DNA mass, measurements of single chromosomes were within the expected range of DNA content. Indeed, single chromosome bridges, lagging chromosomes, and misaligned chromosomes at spindle poles had measured DNA contents within the expected range, as determined based on chromosomal coverage of the genome (Fig 3E). Together, these data validate the use of histone H2B fluorescence intensity as a proxy for DNA content.

## DNA loss corresponds with cell death

We used this method to quantify DNA gain and loss after distinct types of divisions, including multipolar divisions that typically result in cell death. In 72-hour timelapse analysis, DNA content of daughter cells that arose from specific types of divisions was quantified and cell fate was determined. Control cells treated with DMSO and those treated with a subclinical dose of microtubule poison, which typically resulted in 2->2->2 divisions, had DNA contents close to 100%, while centrosome amplification due to dox-inducible Plk4 overexpression produced daughter cells with a wide range of DNA content (Fig 4A). When comparing DNA content between the types of divisions, daughter cells that arose from divisions in which multipolar spindles were maintained throughout mitosis to produce 3$^+$ daughter cells (3$^+$->3$^+$->3$^+$ divisions) had an average of 67% DNA and typically died within the 72-hour observation period (Fig 4B and panels A and B of Fig F in S1 File). 3$^+$->3$^+$->2 divisions produced daughters with a wide range of DNA contents, from approximately 50% to 150%, approximately half of which failed to survive (Fig 4B and Fig F in S1 File). In contrast, daughter cells resulting from 3$^+$->2->2 divisions had 100% DNA content on average and >90% survived, demonstrating that early focusing of multipolar spindles dramatically reduces chromosome missegregation as well as cell death (Fig 4B and Fig F in S1 File).

On average, cells that survived had 100% DNA, while cells that died had lost 20% of their DNA (Fig 4C). To quantify the effect of DNA gain and loss on cell viability, we compared the frequency of death in cells with various amounts of DNA (Fig 4D). Approximately 80% of daughter cells that lost ≥20% DNA died within the 72-hour timelapse analysis. This was substantially higher than the approximately 30% death in cells that gained ≥20% DNA (Fig 4D).

This analysis was performed in diploid cells with wild-type p53 (MCF10A). However, polyploid cells may be resistant to DNA loss because of extra chromosome copies. Additionally,

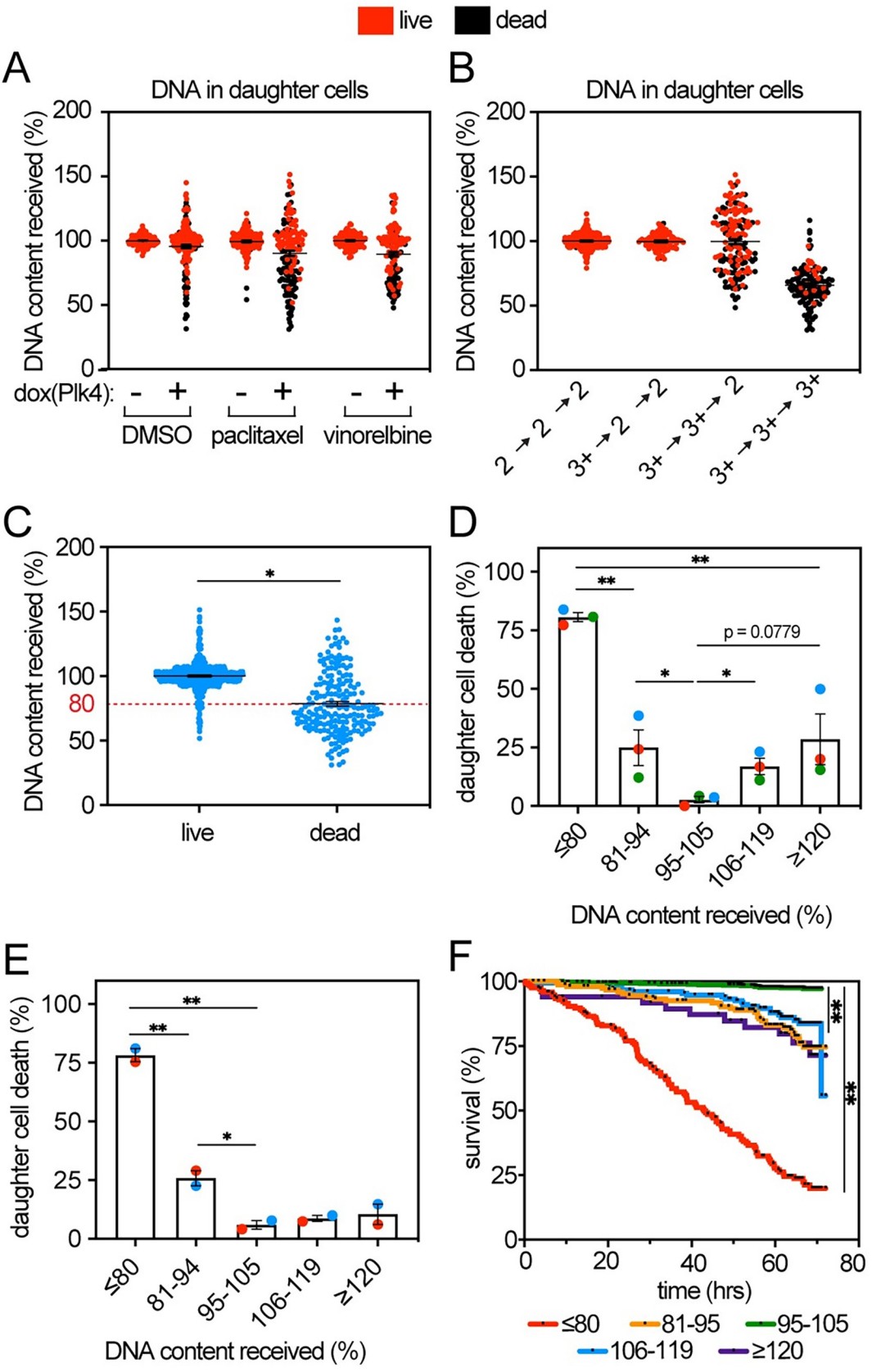

**Fig 4. Frequency and speed of cell death increase with DNA loss.** Data from 72-hour timelapse imaging of Plk4-inducible MCF10A cells stably expressing histone H2B-mNeonGreen and mScarlet-tubulin treated with 1 nM paclitaxel or 1 nM vinorelbine +/− 2 μg/mL dox to induce Plk4. (**A, B**) Quantification of DNA content in daughter cells at early G1 in (**A**) indicated treatment conditions and (**B**) categorized by type of division showing that multipolar divisions increase DNA loss and cell death. $n \geq 50$ cells per replicate in each of 3 biological replicates. (**C**) Quantification of DNA content in daughter cells based on cell fate. $n \geq 60$ cells per condition in each of 3 replicates. Dead cells had an average DNA content of 80% (red line). (**D**) Quantification of cell death in daughter cells, binned by DNA content in MCF10A cells. Loss of $\geq$20% DNA typically results in cell death in <72 hours. $n \geq 40$ cells per category in 3 replicates. Color represents specific replicate. (**E**) Quantification of daughter cell death binned by DNA content in 72-hour timelapse analysis of polyploid, p53 mutant MDA-MB-231 cells. $n \geq 68$ cells per category in 2 replicates. (**F**) Kaplan–Meier survival curve of MCF10A cells by DNA content showing that cell death occurs most rapidly in cells that have lost $\geq$20% DNA content, though it often takes >24 hours. Black points along curves indicate censored cells, which left the field of view or entered a second mitosis. $n$ = 171, 156, 589, 159, and 50 cells for categories $\leq$80, 81–94, 95–105, 106–119, and $\geq$120% DNA, respectively. Unpaired $t$ test was performed on C and D, while Mantel–Cox test was performed on E to determine statistical significance. Bars represent mean +/− SEM. Data used to generate graphs can be found in S1 Data. * indicates $p < 0.05$. ** indicates $p < 0.01$. ns indicates not significant.

though p53 function was not required for cell death in response to high CIN in murine tumors or mouse embryonic fibroblasts [73], whole chromosome missegregation did not activate p53 in human cells [70], and induced whole chromosome aneuploidy can be propagated in p53 proficient cells [75], monosomic cell lines were only recovered in RPE cells deficient in p53 signaling [76], suggesting that p53 signaling may impact viability after DNA loss. We therefore tested whether polyploid (near-triploid) MDA-MB-231 breast cancer cells, which express the R280K mutant of p53, are more resistant to DNA loss. When comparing DNA content with cell survival, polyploid p53-mutant cells behaved very similarly to diploid p53-proficient cells (compare Fig 4D to 4E). Thus, loss of $\geq$20% DNA causes efficient cell death even in p53 deficient, polyploid cells.

Cell death occurred more rapidly in cells that lost $\geq$20% DNA than in cells that gained DNA (Fig 4F). However, even in cells that lost $\geq$20% DNA, death was often delayed and occurred with highly variable timing, ranging from 0.1 to 68 hours after division, with an average of 34 hours (Fig 4F), suggesting that cell death does not occur as a general response to DNA loss but rather due to the absence of specific gene products with independent half lives. Thus, maintenance of multipolar spindles results in high rates of DNA loss and cell lethality and represents an effective mechanism of toxicity for distinct microtubule poisons.

## Symmetric DNA division is more likely to result in successful cytokinesis

Given the importance of successful multipolar division for DNA loss and cell death, we analyzed multipolar divisions to identify factors affecting the likelihood of generating 3$^+$ daughter cells. The vast majority of multipolar divisions were tripolar, while 5$^+$ divisions were rare (Fig 5A). Furthermore, tripolar divisions were significantly more likely to be successful, which we defined as producing the same number of daughters as directions in which the DNA was segregated after anaphase onset (i.e., tripolar divisions producing 3 daughter cells, tetrapolar divisions producing 4 daughters, etc.). Approximately 37% of tripolar divisions produced 3 daughter cells, while the incidence of successful division in cells that entered anaphase with 4 or 5+ poles was <10% (Fig 5B). Multipolar divisions frequently resulted in partial cytokinesis failure, in which tripolar divisions produced 2 daughter cells and cells in which the DNA was divided in 4 or 5+ directions formed 2 to 3 daughters (Fig 5C). We then focused on tripolar divisions because of their frequency and possibility of successful cytokinesis. Based on the DNA content in daughters formed from 3$^+$->3$^+$->3$^+$ versus 3$^+$->3$^+$->2 divisions (Fig 4B), we hypothesized that cells with divisions in which the DNA was partitioned symmetrically in anaphase (i.e., equal fluorescence intensity in each segregating mass within a cell) were more likely

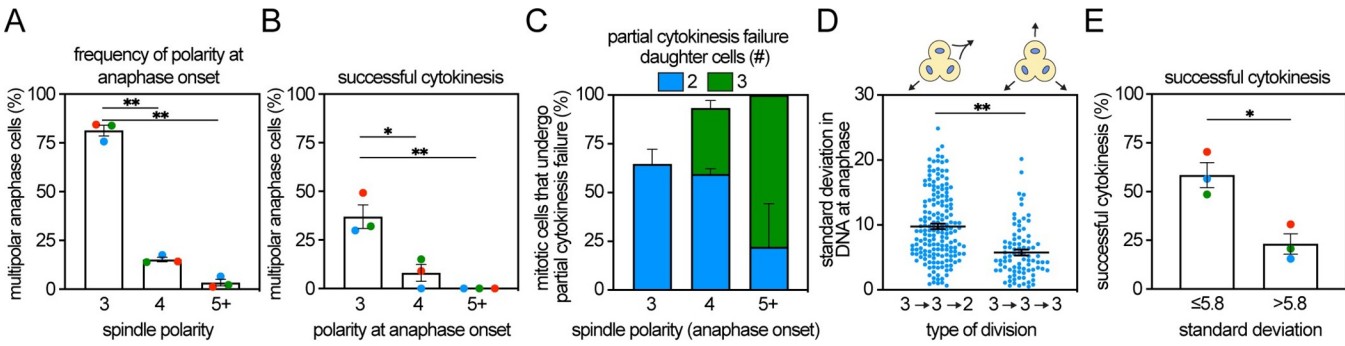

**Fig 5. Symmetric tripolar divisions are more likely to form 3 daughter cells.** (**A**) Quantification of the frequency with which MCF10A cells divide in 3, 4, or 5+ directions at anaphase onset. Analysis was performed from timelapse sessions across all treatment conditions, as described in Fig 4A. Tripolar anaphases are the most common type of multipolar division. $n \geq 77$ cells analyzed in each of 3 independent replicates. (**B**) Frequency of successful cytokinesis in all cells dividing in a multipolar fashion with the indicated spindle polarity at anaphase onset. Percentage shown is the percent of cells with the indicated polarity at anaphase onset that underwent successful cytokinesis. Successful cytokinesis is defined as forming the same number of daughter cells as the number of spindle poles at anaphase onset. (**C**) Quantification of partial cytokinesis failure in cells with the specified multipolar division. $n \geq 10$ cells analyzed per category. (**D**, **E**) In cells that are tripolar at anaphase onset, those that successfully form 3 daughters have a lower standard deviation in daughter cell DNA content than those that exhibit partial cytokinesis failure to form 2 daughter cells. $n \geq 57$ cells per category in each of 3 independent replicates. Unpaired $t$ test was performed to determine statistical significance. Bars represent mean +/− SEM. Data used to generate graphs can be found in S1 Data. * indicates $p < 0.05$. ** indicates $p < 0.01$.

to form 3 daughter cells. To test this, we determined the standard deviation of histone H2B fluorescence in the DNA masses of dividing cells within 5 minutes after anaphase onset as a measure of symmetry. Tripolar divisions that successfully generated 3 daughter cells exhibited more symmetric separation of DNA, with a lower standard deviation (on average, 5.8%) than tripolar divisions that underwent partial cytokinesis failure into 2 daughter cells (Fig 5D). Tripolar divisions with a standard deviation ≤5.8% were almost 3-fold more likely to exhibit successful cytokinesis than divisions with >5.8% standard deviation in the segregating DNA masses (Fig 5E). Taken together, symmetric DNA partitioning in early anaphase is significantly more likely to produce successful cytokinesis and daughter cell death than asymmetric division.

## Loss of an additional 3 to 4 chromosomes confers sensitivity to microtubule poisons, even in cells that focus multipolar spindles

These data support a model in which multipolar divisions are lethal because they increase CIN over a maximally tolerated threshold; early focusing of multipolar spindles causes resistance because it dramatically reduces CIN. According to this model, cells that are resistant to microtubule poisons because of early spindle focusing could be sensitized by addition of another CIN-inducing drug. However, the rate of CIN necessary to sensitize to microtubule poisons is unknown. To better define the threshold of CIN that confers sensitivity to paclitaxel, we generated 2 stable cell lines that induce low rates of missegregation due to dox-inducible expression of either (1) a dominant negative mutant form of TRF2, TRF2(ΔBΔM) [77], which induces chromosome bridges, or (2) shRNA directed against Kif2b, which produces lagging chromosomes [78]. Dox-inducible TRF2(ΔBΔM)-mScarlet expression was confirmed with visualization of mScarlet fluorescence. Since endogenous Kif2b expression is below the lower limit of detection in most human cell types, and Kif2b shRNA was shown to reduce Kif2b mRNA expression in HEK293T cells [79,80], we confirmed successful reduction of Kif2b mRNA expression (panel E of Fig D in S1 File). In the presence of dox, expression of TRF2(ΔBΔM)-mScarlet and Kif2b shRNA predominantly resulted in a single chromosome bridge or lagging chromosome, respectively (panels A and B of Fig G in S1 File). However, these low rates of

missegregation had no effect on viability in combination with paclitaxel (panels C and D of Fig G in S1 File), suggesting one missegregation is insufficient to induce sensitivity.

We previously observed that partial inhibition of the mitotic kinesin CENtromere protein E (CENP-E) with GSK923295 is synergistically lethal with paclitaxel [49]. While high concentrations of GSK923295 cause mitotic arrest, lower doses of the inhibitor permit cells to proceed through mitosis after a delay, similar to partial depletion with siRNA [81,82]. Here, we demonstrate that CENP-E inhibition (CENP-Ei) also results in synergistic lethality with low nM concentrations of vinorelbine (Table C in S1 File). CENP-Ei results in chromosomes that remain chronically misaligned at or near spindle poles (polar chromosomes; Fig 6A) [39,83]. To understand how CENP-Ei sensitizes cells to microtubule poisons, Cal51 cells endogenously labeled with histone H2B-mScarlet and tubulin-eGFP were followed by timelapse microscopy for 48 to 72 hours. Cal51 cells were selected since they proficiently focus multipolar spindles early in mitosis ($3^+$->2->2 divisions) in both paclitaxel and vinorelbine, as evidenced by the decrease in multipolar spindles as cells transition from early to late stages of mitosis (Fig 6B and 6C). This focusing was unaffected by the addition of CENP-Ei (Fig 6B and 6D). The major impact of including CENP-Ei with microtubule poisons was introducing a population of cells that entered anaphase with ≥3 polar chromosomes (Fig 6E), which correlated with a notable increase in cell death (Fig 6F). Importantly, in cells that are resistant to paclitaxel and vinorelbine due to early focusing of multipolar spindles ($3^+$->2->2 divisions), inclusion of CENP-Ei substantially increased the rate of cell death (Fig 6G).

We then quantified DNA content in daughter cells formed after treatment with microtubule poisons +/− CENP-Ei. On average, daughter cells that died after inclusion of CENP-Ei with paclitaxel or vinorelbine had lost approximately 6% of their DNA content (Fig 6H). Concentrating on the cells that are resistant to microtubule agents because they focus multipolar spindles into bipolar spindles early in mitosis ($3^+$->2->2 divisions), we observed that loss of ≥6% DNA was sufficient to significantly increase the incidence of daughter cell death (Fig 6I). To identify how many missegregations were sufficient to induce loss of at least 6% DNA, we quantified DNA content in cells with ≤4 missegregated chromosomes that were clearly separable from the surrounding DNA. Missegregation of 3 or 4 chromatids was sufficient to account for ≥6% DNA in this near-diploid cell line (Fig 6J). Of 10 daughter cells arising from $3^+$->2->2 divisions in which 3 to 4 polar chromosomes were lost, 9 died within the timelapse analysis. Together, these results support the conclusion that increasing CIN by a rate that induces loss of an additional 3 to 4 chromatids per diploid genome is sufficient to sensitize cells that achieve resistance to paclitaxel and vinorelbine through multipolar spindle focusing.

## Discussion

Microtubule-targeting drugs are a mainstay of treatment for various cancers. For decades, these agents were expected to exert their anticancer activity by inducing mitotic arrest in patient tumors. Because of this, substantial drug discovery efforts were directed at developing agents that could arrest cells in mitosis without affecting microtubules. The failure of numerous such agents to exert efficacy in clinical trials was disappointing and caused some to speculate that the efficacy of microtubule poisons is due to effects on interphase microtubules rather than mitosis [45,46]. Others have continued to focus on mitotic arrest as the clinically relevant mechanism of these drugs [84]. It is now clear that the concentration of microtubule poisons achieved in breast cancers has a considerable effect on mitosis, without causing mitotic arrest. Clinically relevant concentrations of diverse microtubule poisons cause an at least transient increase in multipolar spindle poles. Maintenance of multipolar spindles throughout mitosis causes high rates of CIN and sensitivity to the antimicrotubule agent, but focusing of

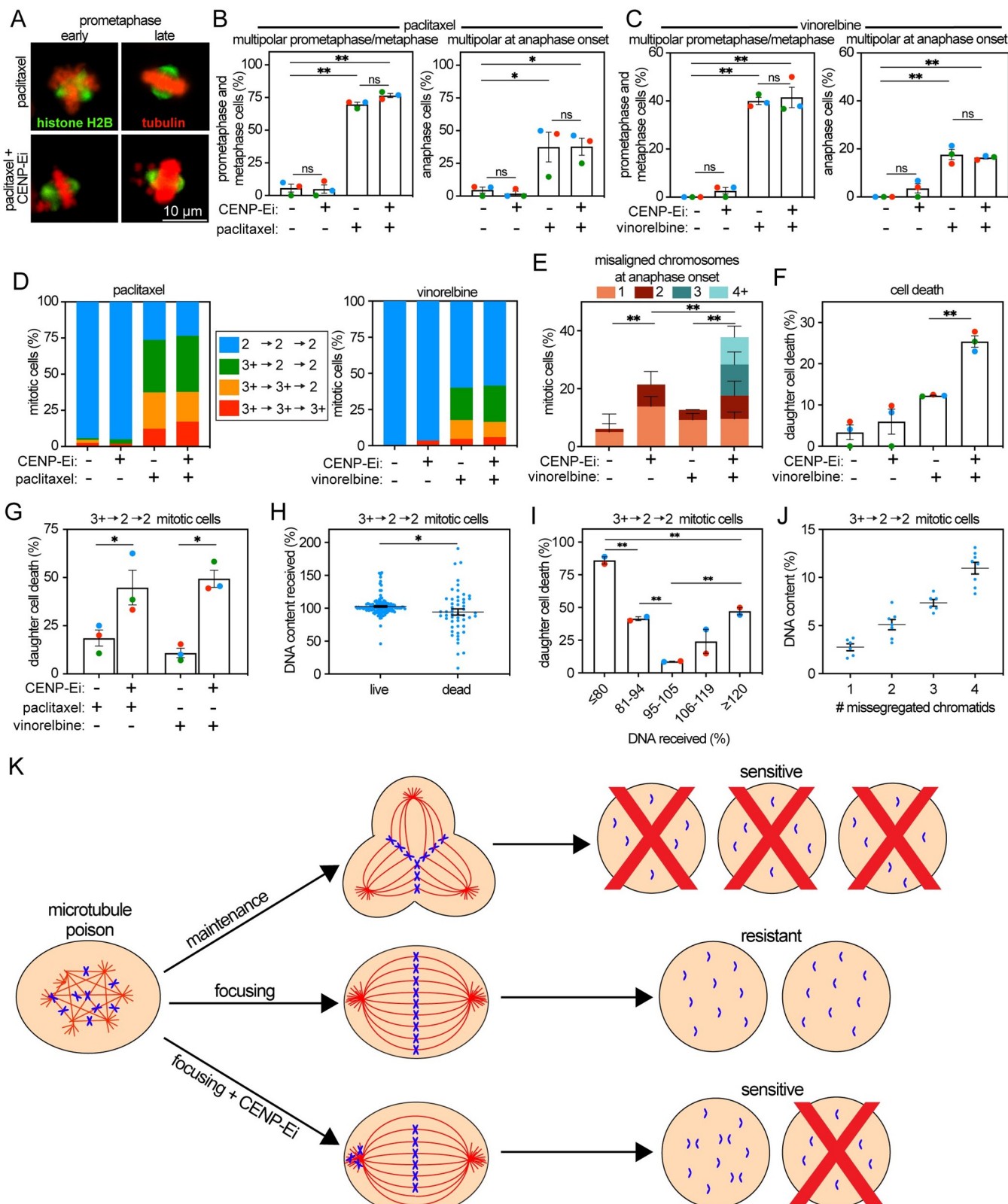

**Fig 6. Loss of 3–4 chromosomes per diploid genome is sufficient to increase sensitivity to microtubule poisons.** Data from 48–72-hour timelapse imaging of Cal51 cells endogenously expressing H2B-mScarlet and tubulin-eGFP treated with vehicle, 5 nM paclitaxel, 4 nM vinorelbine, and/or 50 nM CENP-E

inhibitor GSK923295. (**A**) Representative still frames of multipolar spindles before and after focusing into bipolar spindles. (**B**, **C**) Quantification of multipolar spindles at prometaphase+metaphase and anaphase onset in cells treated with (**B**) paclitaxel or (**C**) vinorelbine showing that CENP-E inhibition does not impair focusing of multipolar spindles. (**D**) Quantification of spindle polarity throughout mitosis in cells treated with paclitaxel (left) or vinorelbine (right) +/− CENP-E inhibition showing that CENP-E activity does not affect spindle polarity. (**E** and **F**) Quantification of misaligned chromosomes (**E**) and daughter cell death (**F**) by indicated treatment showing that CENP-E inhibition substantially increases both (**E**) the missegregation of misaligned chromosomes and (**F**) daughter cell death in cells treated with vinorelbine. (**G**) CENP-E inhibition sensitizes cells that focus multipolar spindles early (3+ -> 2 -> 2 divisions) to microtubule poisons. (**H**) Quantification of DNA content in daughter cells that focus microtubule poison-induced multipolar spindles early in mitosis (3+ -> 2 -> 2 divisions) based on final cell fate. (**I**) Quantification of cell death in daughter cells resulting from early focusing events (3+ -> 2 -> 2 divisions), as defined by DNA content at early G1. (**J**) Quantification of DNA content in indicated number of misaligned chromosomes. (**K**) Model: Microtubule poisons induce multipolar spindles early in mitosis. Though multipolar divisions are lethal, focusing into near-normal bipolar spindles early in mitosis reduces CIN and causes resistance. This resistance can be overcome by (1) preventing focusing or (2) another insult that induces CIN, such as CENP-E inhibition (CENP-Ei). Unpaired $t$ test was performed to determine statistical significance. Color represents specific replicate. Data represent mean +/− SEM. Data used to generate graphs can be found in S1 Data. * indicates $p < 0.05$, ** indicates $p < 0.01$.

multipolar spindles can occur at any point during mitosis. Focusing reduces chromosome missegregation and causes resistance. The timing of focusing dictates the magnitude of these effects, with the earliest focusing resulting in the most profound resistance.

Sensitizing the approximately 50% of cancers that are resistant to microtubule poisons remains an urgent unmet clinical need. Preventing focusing offers promise as a potential therapeutic strategy, but the means to accomplish this pharmacologically are not currently available in animal models or patients. We and others have previously shown that manipulations that induce CIN increase sensitivity to microtubule-targeted agents [49,67,68,85]. Despite this, the minimal rate of chromosome missegregation required to sensitize tumors to these therapies remained unknown. Additionally, the current gold-standard method of measuring CIN and assessing its downstream consequences, timelapse analysis of cells with fluorescent chromosomes and microtubules, can only be used to quantify cells in which each missegregated chromosome forms a discrete, visually separate, mass. To overcome this, we validated histone H2B fluorescence in timelapse microscopy as a method of measuring DNA and quantifying CIN, though it remains unclear whether this method is sufficiently sensitive to detect loss of a single chromosome within a DNA mass. This nondisruptive technique permits quantification of larger numbers of missegregations and identified a direct relationship between CIN and cell fate. Using this technique, we quantified the effects of successful multipolar divisions into $3^+$ daughter cells on DNA content. On average, cells formed from successful multipolar divisions have lost approximately 1/3 of their DNA (Fig 4B). Loss of $\geq$20% of DNA is typically lethal, even in polyploid cells with mutant p53 (Fig 4B,4D and 4E). DNA loss is reduced to <1% of DNA by early focusing, illustrating the damaging consequences of persistent multipolar spindles and the dramatic effect of focusing prior to anaphase onset. Cells with substantial DNA loss are more likely to experience haploinsufficiency and nullisomy, or loss of all copies of a specific chromosome. Nullisomy of any autosome is expected to be lethal. The amount of DNA loss strongly correlates with the duration of cell survival following mitosis, as cells that lost $\geq$20% DNA died at a faster rate than any other group analyzed (Fig 4F). Surprisingly, however, cell death was a slow process—even after loss of $\geq$20% DNA—as these cells required approximately 34 hours on average to die, with a range of 0.1 to 68 hours. A general cellular response to nullisomy or monosomy could be expected to cause cell death with a similar time course, irrespective of the chromosomes involved, but this was not observed. Since cell death can occur in as little as 10 minutes [86], the wide range of timing of cell death suggests that the triggers are variable, karyotype specific, and based on the half life of products encoded by the deficient chromosomes. Determining the mechanism(s) of cell death following high rates of chromosome loss is an important future direction.

Though microtubule poisons used as anticancer agents are all able to induce multipolar spindles, nocodazole and colcemid—which are not used clinically—are not. The difference

between these microtubule poisons in this respect is quite striking, though the reason for the difference is not readily apparent. We considered 4 factors that might be responsible for this difference: binding stoichiometry, reversibility of drug binding, ability to bind to soluble versus polymerized tubulin, and binding site. No single factor appears to be sufficient to account for the difference in inducing multipolarity. Substoichiometric concentrations of nocodazole and clinically effective microtubule poisons suppress microtubule dynamics [21,22,87]. Nocodazole [88] and the vinca alkaloid vinblastine [89] bind to tubulin in a rapidly reversible manner. Nocodazole [90], vinca alkaloids [21], and eribulin [91] bind to soluble tubulin subunits as well as microtubules. Nocodazole [88] and colcemid [92] bind to the colchicine site on tubulin, while paclitaxel, docetaxel, and epothilones bind to the taxane site [93,94], and vinca alkaloids and eribulin bind to the vinca domain [91]. However, colchicine has been reported to potently induce multipolar spindles at certain concentrations [95]. Interestingly, colchicine binding to tubulin is poorly reversible [21,96], while nocodazole [88] and colcemid [97,98] binding is rapidly reversible. Since neither nocodazole nor colcemid induce multipolar spindles, we hypothesize that microtubule poisons that bind the colchicine site of tubulin in a rapidly reversible manner do not induce multipolar spindles.

Identification of a predictive biomarker of response to microtubule poisons is a second urgent unmet need. The importance of multipolar spindle maintenance in dictating response implicates this as a potential contributor to such a biomarker. However, it is not currently possible to predict whether a given multipolar spindle in prometaphase will be focused or maintained. Multipolarity at prometaphase and metaphase rarely predicts cell fate. Maintenance of a multipolar spindle in telophase was more predictive, but the vast majority of mitotic cells in fixed patient biopsies are in early stages of mitosis [49], and it cannot presently be predicted whether they will ultimately be focused or maintained. Our timelapse analysis indicates that cells that enter anaphase on multipolar spindles with symmetrically segregated DNA are more likely to successfully form 3 daughter cells. However, a more complete understanding of the factors that dictate focusing is necessary to predict focusing in patients.

Importantly, pharmacologically increasing CIN improves sensitivity to microtubule poisons, even in cells that are resistant to treatment because they focus multipolar spindles (Fig 6G; [49]). Increasing missegregation by an amount that resulted in the net loss of an additional 3 to 4 chromatids per diploid genome was sufficient for improving response to diverse microtubule poisons. We hypothesize that a proportional increase in DNA loss is sufficient to sensitize polyploid cells (i.e., 4 to 6 chromosomes in a triploid cell). Since chromosome missegregation results in chromosome gain as well as loss, the overall missegregation rate must likely be higher than 3 to 6 chromosomes in order to achieve net loss of this amount of DNA.

Since analysis of histone H2B fluorescence does not distinguish between chromosomes, it is not possible at this point to determine whether the observed increase in cell death was due to nullisomy. Interestingly, previous work has indicated that chromosome missegregation is often nonrandom; in human cells, larger chromosomes tend to be preferentially missegregated [99–102], though it is important to note that chromosome missegregation in these experiments was not due to multipolar divisions. Preferential loss of large chromosomes may increase the likelihood of nullisomy after loss of a relatively small number of chromosomes. Alternatively, it is possible that monosomy is sufficient for cell death. In support of this hypothesis, monosomic RPE1 cells could only be generated in p53-deficient cells [76], though it is unclear whether p53 caused cell death or prevented proliferation in this context. Performing similar timelapse microscopy experiments on cell lines with multiple, specifically labeled chromosomes will provide greater insight into the consequences of monosomy and nullisomy on cell fate.

Though not to the same extent as chromosome loss, chromosome gain also increased cell death. Nullisomy is unlikely to be the cause of death in more than a small fraction of cells with DNA gains. Gains of 1 or 2 chromosomes induce cellular stress including proteotoxic stress, metabolic dependencies, activation of the immune response, and increased DNA damage, which could contribute to cell death [69–71,103–105].

In summary, multiple microtubule drugs—including those traditionally considered to both stabilize and destabilize microtubules—increase multipolar spindles in metastatic breast cancer and relevant models. These drugs exhibit a conserved mechanism of action, in which cells transit mitosis with multipolar spindles and ultimately segregate their chromosomes in anaphase. Maintenance of multipolar spindles throughout mitosis typically results in daughter cell death, while focusing of multipolar spindles into bipolar spindles is a conserved mechanism of resistance to antimicrotubule agents. The resistance caused by focusing can be overcome by increasing CIN with a second pharmacological insult that induces the loss of at least 3 to 4 additional chromatids per diploid genome. Thus, diverse antimicrotubule agents exhibit a conserved anticancer mechanism that involves increasing CIN over a maximally tolerated threshold due to multipolar divisions. These results suggest that future drug discovery efforts aimed at recapitulating the efficacy of antimicrotubule agents should not focus on agents that induce mitotic arrest but instead on CIN-inducing drugs. These drugs have the advantage of exhibiting potential utility as single agents as well as in combination with antimicrotubule therapy.

## Materials and methods

### Microtubule poison study design

Patients who volunteered to participate in this study were enrolled in a prospective trial at the UW Carbone Cancer Center specifying the treatment, biopsy, and analysis plan. The protocol was approved by the UW Health Sciences Institutional Review Board, assigned UWCCC protocol number UW16151, conducted in accordance with the ethical standards established in the 1964 Declaration of Helsinki and registered on clinicaltrials.gov (NCT03393741). Patients were enrolled if they had metastatic or incurable breast cancer, for which antimicrotubule chemotherapy was indicated. Enrolled patients provided written, informed consent. Patients received standard-of-care microtubule poison treatment (either taxane, eribulin, or vinorelbine). There were no major complications from protocol-mandated research biopsy. Response was assessed based on RECIST 1.1 criteria [106].

This is an ongoing study of the mechanism of microtubule poisons in human breast cancer. All patients had metastatic or incurable breast cancer for which antimicrotubule chemotherapy was recommended per standard of care. A research biopsy was obtained approximately 20 hours after start of the first infusion. Follow-up scans were taken every 3 cycles, as per standard of care. Follow-up was discontinued either 2 months after completion of study treatment or upon systemic imaging following therapy completion (whichever was later). The objectives were to measure intratumoral drug levels, to determine their effects on mitosis, chromosomal instability, and cell proliferation, and to correlate these with response to treatment.

### Cell culture

Plk4-inducible MCF10A cell line (a kind gift from Dr. David Pellman [74]) was grown in DMEM/F12 supplemented with 5% (vol/vol) horse serum, 20 ng/mL human EGF, 0.5 mg/mL hydrocortisone, 100 ng/mL cholera toxin, 10 μg/mL insulin, and 50 μg/mL penicillin/streptomycin at 37°C and 5% $CO_2$. Cal51 (DSMZ) and HEK293T cell lines were grown in DMEM supplemented with 10% (vol/vol) FBS, 2 mM L-glutamine, and 50 μg/mL penicillin/

streptomycin at 37˚C and 5% $CO_2$. GSK923295 (AdooQ Bioscience) was dissolved in DMSO and used at 50 nM final concentration. TRF2(ΔBΔM) and Kif2b shRNA cell lines were generated in parental Cal51 cells that were uniformly expressing TetR. Cal51 TetR-expressing cells were transduced with a retrovirus affording stable integration of a puromycin-resistant marker and either tet-inducible TRF2(ΔBΔM)-mScarlet or Kif2b short hairpin RNA (shRNA) constructs. Following puromycin and blasticidin selection, cells were tested for inducible chromosome missegregation events. HEK293T cells were transduced with a retrovirus affording stable integration of a blasticidin resistance marker and Kif2b-NG construct. Following blasticidin and clonal selection, cells were transfected with empty or Kif2b shRNA-containing vectors to test for Kif2b knockdown. For inducible Plk4, TRF2(ΔBΔM), and Kif2b experiments, 2 μg/mL dox was added 48 hours prior to the addition of paclitaxel.

## Cell viability assay

MTT assay was used to assess metabolic cell viability. A total of 20,000 cells were plated into each well of 6-well plates. On indicated day of measurement, cells were incubated with MTT reagent 3-(4,5 dimethylthiazol-2-yl)-2,5diphenyltetrazolium bromide (VWR, 1 mg/mL) for 3 hours at 37˚ C. Media was aspirated after incubation, followed by incubation with 800 μL DMSO and 100 μL Sorenson's glycine buffer (0.1 M glycine, 0.1 M NaCl (pH 10.5) with 0.1 M NaOH) for 10 minutes at 37˚C. Samples were transferred and measured in 96-well plates in triplicate measurements for each condition at 540 nm.

## Immunofluorescence

Patient samples: 5 μm sections of formalin-fixed, paraffin-embedded patient tumor sections were subjected to antigen retrieval in citrate buffer (10 mM sodium citrate, 0.05% Tween-20 (pH 6.75)) at 95 to 100˚C for 30 minutes, serum-blocked overnight, and stained with rabbit anti-NuMA antibody (a kind gift from Duane Compton), mouse anti-γ-tubulin (Sigma, T6557), and pan-cytokeratin (to mark epithelial cells; Novus Biologicals, NBP2-33200AF647) antibodies diluted 1:100 overnight at 4˚C. Alexa Fluor–conjugated secondary antibodies (Invitrogen) were used at 1:200 for 1 hour at room temperature. DNA was stained using DAPI.

Cultured cells: Cells were washed with MicroTubule Stabilizing Buffer (MTSB: 100 mM K-Pipes (pH 6.9), 30% (wt/vol) glycerol, 1 mM EGTA, 1 mM $MgSO_4$), preextracted for 2 minutes with 0.5% Triton X-100 in MTSB at 37˚C, and fixed with 4% formaldehyde in MTSB. All buffers were prewarmed to 37˚C. Cells were blocked in Triton Block [2.5% (vol/vol) FBS, 200 mM glycine, 0.1% Triton X-100 in PBS] for 1 hour at room temperature. Primary antibodies were incubated overnight at 4˚C. Antibodies were diluted in Triton Block. DNA was stained with 1 μg/mL DAPI for 5 minutes, and cells were mounted with Vectashield. Staining was performed with α-tubulin antibody (YL 1/2; 1:1,000 Bio-Rad, MCA77G).

## Microscopy

Images were acquired on a Nikon Ti-E inverted microscope and analyzed with Nikon Elements software. For timelapse analysis, cells were placed in a Tokai Hit stage top incubator with 5% $CO_2$. Five 2-μm z-planes were acquired every 4 to 5 minutes using a 40×/0.75 numerical-aperture objective with perfect focus compensation for 48 to 72 hours. Cells were treated with drugs approximately 1 hour before acquisition unless otherwise noted. For fixed images, maximum projections of 0.2 μm z-stacks collected with a 60×/1.4 or 100×/1.4 numerical aperture objective are shown.

## Statistical analysis

Analysis was performed using GraphPad Prism. Student $t$ tests (two-tailed) and log-rank (Mantel–Cox) tests were used to assess significance. Statistical parameters including the number of cells analyzed and the number of replicates is reported in the respective figure legends.

## Supporting information

**S1 File. Supporting figures and tables. Fig A. Multipolar spindle induction is a conserved mechanism of clinically useful microtubule poisons.** Diverse microtubule poisons are capable of inducing multipolar spindles at low nM doses in (**A**) Cal51 and (**B**) MDA-MB-231 breast cancer cells after 20 hours of treatment. $n \geq 100$ cells in each of 3 biological replicates. Data used to generate graphs can be found in S1 Data. **Fig B. Low concentrations of microtubule poisons do not induce mitotic arrest.** Mitotic index after 20 hours of treatment with the indicated concentrations of microtubule poisons in (**A**) Cal51 and (**B**) MDA-MB-231 cells. $n \geq 295$ cells across 3 independent replicates. Data used to generate graphs can be found in S1 Data. **Fig C. The clinically ineffective microtubule poisons nocodazole and colcemid do not induce multipolar spindles.** Cal51 and MDA-MB-231 cells were treated with the indicated concentrations of nocodazole (**A-D**) or colcemid (**E-H**) for 20 hours and scored for spindle multipolarity (**A, B, E, F**) and mitotic index (**C, D, G, H**). $n = 100$ cells for spindle polarity and 250 cells for mitotic index in each of 3 independent replicates. Data used to generate graphs can be found in S1 Data. **Fig D. Validation of CIN-inducible cell lines.** (**A**, **B**) Representative images of interphase cells with (**A**) normal and (**B**) amplified centriole numbers. Images were acquired from Plk4-inducible MCF10A cells treated with (**A**) water or (**B**) 2 μg/ mL dox for 72 hours. (**C**, **D**) Quantitation of centriole (**C**) amplification and (**D**) number in dox-inducible Plk4 MCF10A cells after 72 hours of dox treatment. $n = 100$ cells in each of 3 biological replicates. (**E**) Relative Kif2b expression 72 hours after transfection with indicated shRNA. Since endogenous Kif2b expression is below the lower limit of detection, and fluorescently labeled exogenous Kif2b has previously been shown to be a suitable surrogate for endogenous Kif2b [107,108], validation of Kif2b depletion was performed in HEK293T cells stably expressing Kif2b-mNeonGreen. Kif2b expression was normalized to GAPDH expression and empty vector. $n = 3$ biological replicates. Unpaired $t$ test was performed to determine statistical significance. Data used to generate graphs can be found in S1 Data. ** indicates $p < 0.01$. **Fig E. Centriole amplification does not induce multipolar divisions in Cal51 breast cancer cells, which are proficient at focusing multipolar spindles.** (**A**, **B**) Quantitation of centriole (**A**) amplification and (**B**) number in dox-inducible Plk4 Cal51 cells after 48 hours dox treatment showing dox inducible centriole amplification. $n = 100$ cells in each of 3 biological replicates. (**C**, **D**) Quantification of multipolar spindles in (**C**) early stages of mitosis (prometaphase and metaphase) and (**D**) late stages of mitosis (anaphase and telophase) in dox-inducible Plk4 Cal51 cells, showing centriole amplification induces a much lower rate of multipolar spindles in Cal51 cells than in MCF10A cells, even in the presence of subclinical paclitaxel (compare to Fig 2A and 2B and S4 Movie). $n = 100$ cells in each of 3 biological replicates. Unpaired $t$ test was performed to determine statistical significance. Data used to generate graphs can be found in S1 Data. * indicates $p < 0.05$. **Fig F. Sustained multipolarity in the presence of microtubule poisons corresponds with cell death.** Data from 72-hour timelapse imaging of Plk4-inducible MCF10A cells stably expressing histone H2B-mNeonGreen and mScarlet-tubulin treated with 1 nM paclitaxel or 1 nM vinorelbine +/− 2 μg/mL dox to induce Plk4. (**A**, **B**) Quantification of DNA content in daughter cells at early G1 categorized by type of division showing that multipolar divisions increase DNA loss and cell death. $n \geq 50$ cells per replicate in each of 3 biological replicates. (**C**, **D**) Quantification of cell death after the specified type of

division showing that persistent multipolar ($3^+$->$3^+$->$3^+$) divisions are the most lethal across both (**C**) paclitaxel and (**D**) vinorelbine. Color represents specific replicate, and bars represent mean +/− SEM. $n \geq 40$ cells per category across 3 biological replicates. Unpaired *t* test was performed to determine statistical significance. Data used to generate graphs can be found in S1 Data. * indicates $p < 0.05$, ns indicates not significant. **Fig G. A single missegregated chromosome is not sufficient to sensitize to paclitaxel.** (**A**) Quantitation of chromosome bridges in dox-inducible TRF2(ΔBΔM)-mScarlet Cal51 cells after 72 hour dox treatment. $n = 50$ cells in each of 3 biological replicates. (**B**) Quantitation of lagging chromosomes in dox-inducible Kif2b shRNA Cal51 cells treated with dox for 72 hours. $n = 50$ cells in each of 3 biological replicates. (**C**, **D**) Dox treatment of dox-inducible TRF2(ΔBΔM)-mScarlet (**C**) or Kif2b shRNA (**D**) Cal51 cells does not increase sensitivity to paclitaxel. Cells were treated with 2 μg/mL dox for 48 hours before treatment with 2.5 nM paclitaxel for the indicated number of days. $n = 3$ biological replicates. Unpaired *t* test was performed to determine statistical significance. Data used to generate graphs can be found in S1 Data. * indicates $p < 0.05$. **Table A. Metastatic microtubule poison trial patient characteristics.** ER, estrogen receptor. PR, progesterone receptor. HER2, human epidermal growth factor receptor 2. All patients in this study had metastatic spread. Patient response was determined by review of imaging reports and RECIST 1.1 criteria [106]. Response information was not available for Patient 1 because they only received one dose of eribulin. **Table B. Intratumoral eribulin concentration.** Eribulin concentration was measured by HPLC analysis 20 hours after the first dose of eribulin in plasma and 2 tumor cores from a single patient. Eribulin was quantified assuming a tumor density of $1g/cm^3$. NA, not applicable. **Table C. CENP-E inhibition synergizes with low doses of vinorelbine.** Chou-Talalay non-constant ratio synergy testing of vinorelbine with the CENP-E inhibitor GSK923295 in Cal51 cells. Combination index (CI) = 1 indicates an additive response, CI > 1 an antagonistic one, and CI < 1 synergistic. m = kinetic order of single drug curves, Dm = $IC_{50}$, r = linear correlation coefficient for median affect plot, NA, not applicable. (PDF)

**S1 Movie. 2->2->2 division.** Timelapse analysis of Plk4-inducible MCF10A cell stably expressing histone H2B-mNeonGreen and mScarlet-tubulin undergoing a normal division in which the mitotic spindle has 2 poles in prometaphase, 2 poles at anaphase onset, and forms 2 daughter cells (designated 2->2->2 division).
(MP4)

**S2 Movie. $3^+$->2->2 division.** Timelapse analysis of Plk4-inducible MCF10A cell stably expressing histone H2B-mNeonGreen and mScarlet-tubulin with a multipolar ($3^+$ poles) mitotic spindle in prometaphase that focuses to a bipolar spindle (2 poles) at anaphase onset and forms 2 daughter cells ($3^+$->2->2 division).
(MP4)

**S3 Movie. $3^+$->$3^+$->2 division.** Timelapse analysis of Plk4-inducible MCF10A cell stably expressing histone H2B-mNeonGreen and mScarlet-tubulin with a multipolar ($3^+$ poles) mitotic spindle in prometaphase that remains multipolar ($3^+$ poles) at anaphase onset but forms only 2 daughter cells ($3^+$->$3^+$->2 division).
(MP4)

**S4 Movie. $3^+$->$3^+$->$3^+$ division.** Timelapse analysis of Plk4-inducible MCF10A cell stably expressing histone H2B-mNeonGreen and mScarlet-tubulin with a multipolar ($3^+$ poles) mitotic spindle in prometaphase that remains multipolar ($3^+$ poles) at anaphase onset and forms 3 daughter cells ($3^+$->$3^+$->$3^+$ division).
(MP4)

**S1 Data. Underlying data for Figs 1–6 and Figs A-G in S1 File.**
(XLSX)

## Acknowledgments

We thank S. Godinho and D. Pellman for the inducible Plk4 MCF10A cell line, D. Compton for NuMA antibody, T. Kinoshita in the Translational Research Initiatives in Pathology (TRIP) lab for histology services, our patients for their participation in this research, and members of the Weaver, Burkard, and Suzuki laboratories for insightful discussions.

## Author Contributions

**Conceptualization:** Amber S. Zhou, John B. Tucker, Andrew R. Lynch, Mark E. Burkard, Beth A. Weaver.

**Formal analysis:** Amber S. Zhou, Christina M. Scribano, Caleb L. Carlsen.

**Funding acquisition:** Mark E. Burkard, Beth A. Weaver.

**Investigation:** Amber S. Zhou, John B. Tucker, Christina M. Scribano, Caleb L. Carlsen, Sophia T. Pop-Vicas, Srishrika M. Pattaswamy.

**Methodology:** Amber S. Zhou.

**Supervision:** Beth A. Weaver.

**Writing – original draft:** Amber S. Zhou, Beth A. Weaver.

**Writing – review & editing:** Amber S. Zhou, John B. Tucker, Christina M. Scribano, Andrew R. Lynch, Caleb L. Carlsen, Sophia T. Pop-Vicas, Srishrika M. Pattaswamy, Mark E. Burkard, Beth A. Weaver.

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
