## [Editor Report · Decision Letter 0]

13 Apr 2023

Dear Dr Weaver, 

Thank you for submitting your manuscript entitled "Chromosome missegregation on multipolar spindles is a conserved mechanism of microtubule poisons" for consideration as a Research Article by PLOS Biology. Please accept my apologies for the delay in getting back to you as we consulted with an academic editor about your submission.

Your manuscript has now been evaluated by the PLOS Biology editorial staff, as well as by an academic editor with relevant expertise, and I am writing to let you know that we would like to send your submission out for external peer review.

Once your full submission is complete, your paper will undergo a series of checks in preparation for peer review. After your manuscript has passed the checks it will be sent out for review. To provide the metadata for your submission, please Login to Editorial Manager (https://www.editorialmanager.com/pbiology) within two working days, i.e. by Apr 15 2023 11:59PM.

Kind regards,

Richard

Richard Hodge, PhD

Associate Editor, PLOS Biology

rhodge@plos.org

PLOS

---

## [Decision Letter · Decision Letter 1]

2 Jun 2023

Dear Dr Weaver,

Thank you for your patience while your manuscript "Chromosome missegregation on multipolar spindles is a conserved mechanism of microtubule poisons" was peer-reviewed at PLOS Biology. Please accept my apologies for the long delays that you have experienced during the peer review process. Your manuscript has now been evaluated by the PLOS Biology editors, an Academic Editor with relevant expertise, and by three independent reviewers. 

In light of the reviews, which you will find at the end of this email, we would like to invite you to revise the work to thoroughly address the reviewers' reports.

As you will see, the reviewers think the study is interesting and the findings significant for the field. Reviewers #1 and #2 note that several reporting details should be added for the figures and that controls are needed to validate Plk4 overexpression or Kif2B knockdown. In addition, Reviewer #3 asks that additional experiments are included to show a direct correlation between multipolarity and cell death, as well as including additional discussions and clarifications in the manuscript text.

Given the extent of revision needed, we cannot make a decision about publication until we have seen the revised manuscript and your response to the reviewers' comments. Your revised manuscript is likely to be sent for further evaluation by all or a subset of the reviewers.

**IMPORTANT - SUBMITTING YOUR REVISION**

*Re-submission Checklist*

*Published Peer Review*

*PLOS Data Policy*

*Blot and Gel Data Policy*

Sincerely,

Richard

Richard Hodge, PhD

rhodge@plos.org

REVIEWS:

Reviewer #1 (Claire E Walczak, signs review): Anti-microtubule agent are powerful therapeutics used as front-line treatments in a number of cancers. They have been historically thought to act by arresting cells in mitosis, leading to apoptosis. However, groundbreaking work by the Weaver lab showed that in breast cancer, paclitaxel did not induce mitotic arrest, but rather cells had to pass through mitosis leading to chromosomal instability (CIN), aneuploidy and the formation of micronuclei. This finding suggests that drugs that halt the cell cycle may not be therapeutically valuable, but instead that targeting CIN may be the key mechanism of action.

In the present study, the authors show that other clinically used anti-microtubule agents also generate CIN, leading to cell death as a conserved mechanism of action. Generation of CIN was only observed in the anti-microtubule agents that were used clinically and did not correlate with microtubule-targeting drugs, such as nocodazole, that depolymerize microtubules and are not used therapeutically. They also develop a new method to quantify CIN, which will be a valuable addition to the field.

Overall, the findings in this paper are highly significant both for the cell biology community as well as the cancer biology field, as it is paradigm shifting in thinking about the mechanistic action of anti-microtubule agents. The data is strong and convincing of the points made. I have a few small suggestions for improvement of the paper before publication.

1. Throughout the manuscript, the authors need to clearly specify the number of samples analyzed in terms of patient samples, cells, number of replicates and which statistical tests were used. 

2. There are no controls to validate the overexpression of Plk4 or the knockdown of Kif2B.

3. In the data in Figure 3, they clearly show that the assay can detect a single chromosome mis-segregation event, but I am curious if the assay is sensitive enough to detect loss of a chromosome in the DNA mass. Also- in Figure 3D, five chromosome pairs is an insufficient number of samples to analyze. If the authors did three independent repeats, that is only one to two chromosome pairs per experiment. 

4. In Figure 4E, it would be helpful if the authors chose a different color scheme for the Kaplan Meier curves; they are difficult to read.

5. I am confused by the data using the Cenp-E inhibitor, my recollection is that Cenp-E inhibition leads to a sustained mitotic arrest before failure in checkpoint inhibition. 

Reviewer #2: The paper provides an interesting incremental advance to the field's understanding of microtubule poisons, in particular demonstrating a common mechanism of action across microtubule stabilisers and destabilisers in inducing CIN via multipolarity. They also provide a useful methodology for quantifying CIN resulting from multipolarity, and use it to demonstrate that DNA loss is most detrimental to the cells in their study. In agreement with their previous work, they find that spindle refocussing reduces CIN, providing resistance to microtubule poisons, and that refocussing can be driven by asymmetrical spread of DNA across the poles. Finally, they find that using a second drug to drive misalignment can result in daughter cell death despite spindle refocussing. Overall, the study is well done suitable for publication. Below are a number of minor comments designed to help improve the presentation for the reader.

Figure 1

Need more clinical information re clinical study: How many patients were included in the study? How many received each treatment and how many patients contributed to Fig.1 panels F and G (the legend currently only gives numbers of cells, are these from one or multiple patient samples?)

Not sure what they mean by a "clinical touchstone?"

Fig1F in the key the triangle is labelled "C1D2". No clear. Label as "post-treatment"

Figure 2

Legend states "prometaphase" but text states "prometaphase and metaphase".

Not clear if comparisons without stats are non-significant i.e. is the increase in multipolarity with PLK4 induction alone in A/B non-significant? Later plots do use "NS" to label non-significant comparisons (maybe should be consistent).

Why are the stats comparisons across different treatments missing from A (but included in B and C)?

Consider discussing the increase in multipolar spindles with PLK4 induction alone (Panels A, B and E) and how this contributes to the impact of the microtubule poisons + PLK4 OE (i.e. do the combinations have significantly more multipolarity than PLK4 OE alone early in mitosis or just late mitosis).

F and G: The frequency of death for 2,2,2 divisions does not seem consistent between these two panels. Are the data for 2,2,2 and 3,2,2 switched in Panel G?

H-I: Range of number of cells counted is quite large (22-246); could break this down by condition to allow the reader to understand how many cells contributed to each condition.

Legend doesn't state what the bars on the bar graphs represent.

Legend should state that the colours represent replicates. Or make all replicates black as the colouring of the replicates gets confusing alongside the colouring in panel E.

Figure 3

Panel C: Not sure why also showing 25 and 50 mins before NEBD? These timepoints are not mentioned in the text.

Legend doesn't say what the bars on the plots represent?

Not clear what cell line is used in C-E.

Figure 4

Legend doesn't say what the bars on the plots represent.

Figure 5

Legend and text don't describe the cell line or the treatment used to generate the multipolar divisions for the whole figure.

Panel A: Is it the % of the "multipolar anaphase cells" or the % of all anaphase cells (think the former i.e. of the multipolar they are mostly tripolar)? Should make clearer.

Panel B: Should the y-axis have the same label as the y-axis in panel A?

Legend doesn't say what the bars on the plots represent.

Reviewer #3: In their previous papers, the Weaver lab has shown that low doses of microtubule stabilising drug paclitaxel on cultured cells that mimic clinically relevant doses, apparently cause cell death through promoting multipolar divisions and aneuploidy, rather than the prolonged mitotic arrest seen in higher doses. They also related cell death to the capacity for cancer cells to focus/cluster their multipolar spindles into pseudo-bipolar spindles; the earlier they did this in mitosis, the better their chance of successful cell division and surviving. Using a HSET inhibitor or Plk4 overexpression increased cell death through multipolar divisions, whilst increasing chromosomal instability through a CENPEi synergistically increased cell death with paclitaxel, and inherently high level of CIN in tumours was a good prognosticator for taxane treatment efficacy. 

In this paper, Zhou et al have now expanded this work to include both several different stabilisers and also microtubule destabilising drugs. They confirmed that both types of drugs (with the notable exception of nocodazole) alone promote multipolar divisions and cell death, and that this effect is exacerbated with Plk4-related centrosome amplification and again was related to whether cells clustered their multipolar into bipolar spindles early enough in mitosis to avoid a multipolar division. They used histones tagged with fluorescent markers to track DNA content in live cell division, and assessed that cells that tended to lose over 20% of their genome during aberrant divisions were more likely to die and to die faster, than cells that gained 20% extra DNA. Instability generated by Kif2b shRNA or telomere dysfunction which generally resulted in only lagging chromosomes or bridges was insufficient to sensitise to paclitaxel. Furthermore, CENPEi was useful for promoting extra cell death in cell line Cal51 which was resistant to paclitaxel due to effective spindle focusing, by working synergistically. 

Overall, major conclusions of their study are that (i) low doses of either microtubule stabilising, or destabilising agents, that are representative of clinically achieved intratumoural levels, induce multipolar spindles in breast cancer cells, (ii) cells can be sensitised to low doses of microtubule disrupting agents by either increasing centrosome numbers (using Plk4 overexpression in MCF10A normal breast epithelial cells) or (iii) increasing chromosome mis-segregation (using CENP-E inhibition in Cal51 breast cancer cells)).

The paper is in general well presented and holds interest for the readership of Plos Biology. I have a few concerns that would need to be addressed in order to strengthen the authors' current conclusions:

Major points:

1. At present the manuscript seems to imply that the action of microtubule (MT) poisons in causing multipolar spindles is the cause of anti-tumour activity (presumably via cell death). For example, they state in the introduction "Timelapse analysis reveals that diverse microtubule poisons induce a mitotic spindle that starts out multipolar in early mitosis. In cases in which the multipolar spindle is sustained, it results in high CIN and cell death". However the data do not directly show this at present. In Figure 1 they do not correlate multipolar spindle incidence with subsequent cell death. In Figure 6 they use Cal51 cells and 5nM paclitaxel (stabiliser) or vinorelbine (destabiliser) and follow cell death rates, but there are a couple of issues with this data as it is currently presented. First, there seems to be a bit of a mismatch between the rates of multipolar spindles in Cal51 with these two agents between Figure 1 and 6. Cal51 only reaches about 50% multipolar spindles even at 10 nM in Figure 1, but in Figure 6 they show ~70% multipolar spindles in prometaphase at 5 nM paclitaxel for example. Can they comment on this? Or show the data for 5 nM in Figure 1 too? Second, they show the overall rates of cell death after paclitaxel and vinorelbine, but do not correlate with the incidence of the four categories of spindle multipolarity (2-2-2, etc). Instead, they analyse only the 3-2-2- category for cell death. They need to show the direct correlation between multipolarity (either transient or sustained) and cell death to support their statement in the introduction that ""Timelapse analysis reveals that diverse microtubule poisons induce a mitotic spindle that starts out multipolar in early mitosis. In cases in which the multipolar spindle is sustained, it results in high CIN and cell death".

2. All the experiments showing sensitisation to multipolar spindle formation due to centrosome amplification are done in MCF10A cells, rather than in a cancer cell line with 'natural' centrosome amplification (CA). They only tested the synergy between MT poisons and CA using forced CA in normal breast epithelial cell line MCF10A. Why didn't they use a breast cancer line with CA to investigate this? They could have correlated the number of centrioles and proportion of time in prometaphase, anaphase with multipolar spindles etc with cell death. Modelling this in MCF10A normal cells I am not sure how they link this to patients? Do they propose that natural CA would synergise with taxol treatment? They could discuss this more.

3. They present nice data carefully quantifying the levels of alteration in genomic content in daughter cells that originated after a 3-2-2 division, and conclude that loss of genomic material over 20% is highly likely to induce subsequent cell death. Both of the cell lines used to test this are diploid, or near diploid, whereas many cancer cell lines exhibit higher ploidy. MCF10A is a non-transformed breast epithelium line, and Cal51 is also a diploid triple negative cell line (very unusual). Therefore it is likely that most cancer cell lines would have to lose significantly more genomic material in order to be susceptible to nullisomy or monosomy. Either additional cell lines with higher ploidy more representative of breast cancer karyotypes should be tested in this assay, or their conclusions should be clearly caveated by explaining the genomic content of the cells used in the study. Also the authors need to state what is the p53 status of both these lines. P53 status is known to be a large factor in whether diploid cells tolerate the gain, or loss of chromosomes. If they would like to propose that tumour cells could be sensitised to microtubule disrupting agents by increasing CIN then they would need to repeat experiments with more representative cancer cell lines that carry non-diploid karyotypes (and/or defective p53 signalling), to show that the proposed strategy of sensitising to MT agents by elevating CIN by a critical threshold of 3-4 chromosomes mis-segregating could be generally applicable to cancer treatment strategy. E.g. the MDA-MB-231 line is reported to have a triploid genome content which might be expected to easily buffer against the loss of 20% of the genome.

4. In discussion:"it is not possible at this point to determine whether the observed increase in cell death was due to nullisomy. Interestingly, previous work has indicated that chromosome missegregation is often non-random; in human cells, larger chromosomes tend to be preferentially missegregated [80- 83]. This may increase the likelihood of nullisomy after loss of a relatively small number of chromosomes". As far as I am aware no one has yet tested the identities of chromosomes mis-segregated more frequently after multipolar mitosis so it is not clear if larger chromosomes would be more highly mis-segregated in this situation. In my opinion, the more likely explanation is monosomy - which has been shown to be a high barrier to cell fitness, and difficult or impossible to obtain without p53 mutation (work from the Storchova lab). Could they comment on this more extensively in the discussion?

5. In Figure 1 they detect increases in clinical tumour specimens of multipolar spindles after treatment with single agents. Their schematic indicates that the intratumoural concentration was also determined - can they show this data (sorry if I missed it somewhere) and discuss how this matched with the doses used in their experiments to monitor multipolar spindle formation?

Minor points:

1. Figure 1a: Why do they show images of cells treated with 5nM but that is not one of the concentrations they show data for?

2. Figure 3 might be better as supplemental data?

3. Figure 5. Legend could state cell type used

4. Sentence "Cell death occurred more rapidly in cells with ≥20% DNA loss than in cells that gained DNA, though even in these cells death often occurred relatively slowly, requiring >24 hours (Figure 4e)." is not very clear

5. It is very striking and interesting that nocodazole does not cause this effect. Can they speculate in the discussion more on their thoughts on this?

6. Would be easier to understand and compare the data if S1 and S2 were merged (as they are in S3). 

7. A small box listing the drugs and whether they are an MT stabilisier or destabiliser would be helpful in Figure 1. 

8. Figure 2 - why is no significance shown compared to DMSO? Data looks significant. Is it just because box is getting "too busy" with too many stars?

9. Graphs 2G/H/I feel redundant. Could this be streamlined? The authors comment that it is interesting that cell death takes 34hrs. Why is that interesting? Can they expand on that.

10. Figure 3: the authors said the DNA contents are in the "expected range." It seems to be roughly 3-4%. Can they clarify what they expected and how they calculate this in the text.

---

## [Editor Report · Decision Letter 2]

8 Sep 2023

Dear Dr Weaver,

Thank you for your patience while we considered your revised manuscript "Chromosome missegregation on multipolar spindles is a conserved mechanism of microtubule poisons" for publication as a Research Article at PLOS Biology. This revised version of your manuscript has been evaluated by the PLOS Biology editors and the Academic Editor.

Based on our Academic Editor's assessment of your revision, I am pleased to say that we are likely to accept this manuscript for publication, provided you satisfactorily address the following data and other policy-related requests that I have provided below (A-D):

(A) We would like to suggest the following modification to the title: 

“Diverse microtubule-targeted anticancer agents kill cells by inducing chromosome missegregation on multipolar spindles”

(B) Please also ensure that each of the relevant figure legends (both main and supplementary) in your manuscript include information on where the underlying data can be found, and ensure your supplemental data file/s has a legend.

(C) Please ensure that your Data Statement in the submission system accurately describes where your data can be found and is in final format, as it will be published as written there. 

(D) Please note that per journal policy, the model system/species studied should be clearly stated in the abstract of your manuscript.

We expect to receive your revised manuscript within two weeks. 

*Published Peer Review History*

*Press*

Sincerely,

Richard

Richard Hodge, PhD

rhodge@plos.org

PLOS

---

## [Editor Report · Decision Letter 3]

18 Sep 2023

Dear Dr Weaver,

On behalf of my colleagues and the Academic Editor, Jonathon Pines, I am pleased to say that we can accept your manuscript for publication, provided you address any remaining formatting and reporting issues. These will be detailed in an email you should receive within 2-3 business days from our colleagues in the journal operations team; no action is required from you until then. Please note that we will not be able to formally accept your manuscript and schedule it for publication until you have completed any requested changes.

PRESS

Best wishes, 

Richard

Richard Hodge, PhD

rhodge@plos.org

PLOS
